# A Study of Condensates Collected during the Fermentation of Grape Must

Jakub Humaj ⬤, Mojmir Baron ⬤, Michal Kumsta, Jiri Sochor *⬤ and Pavel Pavlousek

Department Viticulture & Enology, Mendel University in Brno, Valticka 337, 69144 Lednice, Czech Republic; xhumaj@mendelu.cz (J.H.); mojmir.baron@mendelu.cz (M.B.); michal.kumsta@mendelu.cz (M.K.); pavel.pavlousek@mendelu.cz (P.P.)
* Correspondence: jiri.sochor@mendelu.cz; Tel.: +420-1910267603

**Abstract:** This article deals with the analysis of the condensates which are formed from fermentation gases during the fermentation of grape must. The experiment was divided into two parts. In the first part, the evolution of the individual volatiles was monitored throughout the whole fermentation process of the Riesling variety. In the second part, the condensates from three different grape varieties (Riesling, Merlot, Sauvignon blanc) were investigated and the total content of the selected volatile substances was measured at the end of the fermentation. Attention was focused on the measurements using a GC-MS (gas chromatography-mass spectrometry) for the volatile substances: isoamyl alcohol, isobutyl alcohol, 1-propanol, ethyl acetate, ethyl hexanoate, ethyl octanoate, ethyl decanoate, acetaldehyde, acetic acid, and acetoin. In addition, changes in the alcohol content of the condensate, with respect to the fermentation phase, were analysed. From the results of part 1, the quantity of the substances under investigation produced during fermentation was determined. The highest concentration of flavour compounds was during the fourth and fifth days of fermentation. The most dominant substance was isoamyl alcohol with a concentration of 1267 mg$^{-1}$. The results of part 2 led to a comparison of the overall profile of volatiles between the varieties. The results showed that the condensates have both a high content of volatile substances and of alcohol. It was also shown that the Sauvignon blanc variant had the highest number of volatile compounds in the representation. The Merlot and Riesling variants were very similar. This product has an exceptionally high potential for further use in the wine or food industry.

**Keywords:** grape must; fermentation; condensates

## 1. Introduction

One of the most important and specific characteristics of wine is its volatile profile. It consists of various chemical compounds and their interactions with each other. The aromatic profile, generally considered to be one of the most unique characteristics of wine, comes from the concentrations of various volatiles and their sensory interactions [1]. It guides the winemaker in the selection of the correct grape varieties and the choice of a suitable processing method and temperature. Huge amounts of other substances are formed during alcoholic fermentation. Many of these are released into the atmosphere along with carbon dioxide; others remain in the fermentation medium. The final quantity of volatiles is governed by the removal of carbon dioxide ($CO_2$) [2]. The rapid release of a huge amount of carbon dioxide (up to 40 L/L of must) during vigorous fermentation leads to the continuous breakdown of volatiles which has consequences for the profile of volatiles in the wine. These substances, together with water vapour and ethanol, form the fermentation gases. Among these gaseous substances, volatile substances which include higher alcohols, terpene esters, and many others play a very essential role. The highest levels of volatile substances found were ethyl acetate, isoamyl acetate, and ethyl hexanoate, which give the gas a fruity aroma [3,4]. The production of the aforementioned volatile

substances plays a very important role in the final quality of the wine. It is important to add that if the concentration of higher alcohols is too high, the aromatic expression of the wine may be negative. This is not the case with esters, where these volatile substances contribute to the aromatic profile of the young wine. The main components of the 'fruity' aroma are considered to be 3-methyl-1-butyl acetate, hexyl acetate, and 2-ethyl hexanoate [5,6].

The problem of volatile stripping increases exponentially with temperature, which is consistent with the pressure curves, so winemakers strive for low fermentation temperatures. However, this slows and may even inadvertently stop enzyme kinetics [7,8]. Last but not least, various parameters are adjusted to influence the finished wine and achieve the best possible results [9,10].

The mix of the fermentation gases produced during grape must fermentation does not yet have a further use. The carbon dioxide that is produced, together with other volatiles and alcohol, is freely released into the atmosphere during fermentation without any further use. This mix of fermentation gases could be captured and compressed, using a compression unit, into a pressure vessel for storage. Most importantly, the capture of the fermentation gases can also be used to produce a condensate of these gases. This condensate contains a large amount of alcohol and a high concentration of aromatics.

Partly due to the very low partial pressures of these compounds, their dew point in the fermentation gas must be extremely low. The lowest technically acceptable temperature for this condensation process is $-78\,°C$. Below this temperature, the carbon dioxide would sublime. Guerrini [11] also introduced such a process. In all these cases, the fermentation gas was cooled to a temperature of $-50\,°C$ [12].

The aim of this work is to analyse the selected volatile and other substances in the condensate formed during grape must fermentation. For these purposes, a system has been developed to produce the condensate. The results of the research will help to provide an understanding of the future practical uses of these condensates.

## 2. Materials and Methods

For clarity, the condensate research program was divided into two main experiments. Part 1 was an investigation into the composition of the condensate during grape must fermentation. The Riesling grape variety was used in the experiment. Part 2 was devoted to a comparison of the measured quantities of various volatile substances from three different varieties, Riesling, Merlot, and Sauvignon blanc. The total volatile content produced by each variety throughout the whole fermentation process was measured.

### 2.1. Condensation

The fermentation tank (A) is a closed vessel of a certain volume, the fermentation gases exit the tank via pipe (B), and the gas in this pipe was cooled to a temperature of $-18\,°C$. The pipe led to a vessel (C) which was enclosed in a cooled environment (D) held at a temperature of $-18\,°C$ (Figure 1). The condensate formed in this vessel and was collected for subsequent analysis.

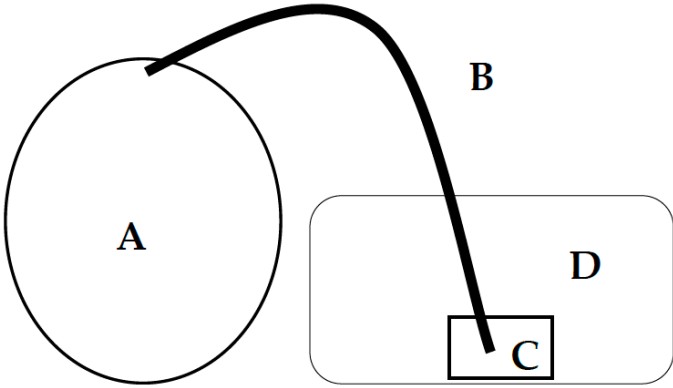

**Figure 1.** Schematic diagram for the collection of condensates.

## 2.2. Design of Individual Experiments

The grapes used for the research were from two villages. Namely Jasová (specific location of the vineyard: 47.992893, 18.432317) and from the village Dubník (location of the vineyard: 47.951183, 18.415953). In both cases, the grapes were treated with non-certified biological protection, i.e., only biological means were used but the vineyard is not certified organic. As far as the soil is concerned, the vineyard is located on sandy loam soil. The vineyards are separated by a relatively small distance of up to 7 km, so the growing conditions throughout the year were similar. The grapes were harvested by hand in good health. They were then transported to the processing site and were crushed, de-stemmed, and pressed. The must was then racked, and fermentation was started using a precise volume of must with a pure yeast culture. This is a neutral yeast which was used to prevent any influence of the yeast on the results to allow a comparison of the varieties. The fermentation temperature was set at 19 °C for all variants. This temperature was chosen to increase the aromatic content of the fermentation gases.

### 2.2.1. Investigation into the Evolution of the Composition of Condensates during Fermentation

For this investigation, the Riesling variety was chosen. The parameters of the grapes are given in Table 1. The grapes were processed and pressed. The must was naturally clarified through sedimentation, and Nutriferm advance (Enartis, San Martino, Italy) was added at a concentration of 20 g·hL$^{-1}$. The must was fermented with a pure culture of Oenoferm Elegance yeast (Erbslöh, Geisenheim, Germany). The fermentation temperature was set to 19 °C. 48 h after yeast inoculation, the container of fermenting must was connected to the condensate harvesting system. This is listed as Day 1 in the results. From this point onwards, the container used to catch the condensate was replaced every 24 h until the end of fermentation.

**Table 1.** Riesling variety (Dubník, Slovakia) parameters.

| Physicochemical Parameters | Sugar | Titrable Acids | pH |
|---|---|---|---|
| Riesling | 22 °BX | 7.7 g·L$^{-1}$ | 3.1 |

### 2.2.2. An Analysis of the Total Condensate

The procedure for the preparation of the must for this experiment was exactly the same as the previous experiment. For this experiment, Riesling was chosen as the representative of the less aromatic varieties. In addition, an overall analysis of Sauvignon blanc was carried out as the representative of the aromatic varieties, and finally, Merlot was chosen to provide results for the analysis of the fermentation condensate of a rosé wine. The grape parameters for each variety are shown in Table 2. The grapes were processed and pressed to produce the must. The must was naturally clarified through sedimentation, and Nutriferm advance (Enartis, San Martino, Italy) was added at a concentration of 20 g·hL$^{-1}$. The must was fermented using a pure yeast culture. The yeast used for the Riesling variety was Oenoferm Elegance (Erbslöh, Geisenheim, Germany), for the Sauvignon blanc variety EnartisFerm ES181 (Enartis, San Martino, Italy), and for the Merlot variety Tipico F3 (Erbslöh, Geisenheim, Germany). The fermentation temperature was set to 19 °C. Immediately after the start of fermentation, i.e., 48 h after inoculation, each fermentation medium was connected to its own condensate collection system. The condensate from each fermentation system was collected in its own collection vessel for the whole of the fermentation process. Subsequently, the collected condensate was subject to analysis.

**Table 2.** Parameters of the three grape varieties.

| Physicochemical Parameters | Sugar | Titrable Acids | pH | Village |
|---|---|---|---|---|
| Riesling | 21 °BX | 7.3 g·L$^{-1}$ | 3.1 | Jasova (Slovakia) |
| Sauvignon blanc | 25.5 °BX | 6.7 g·L$^{-1}$ | 3.27 | Dubník (Slovakia) |
| Merlot | 21.5 °BX | 7.4 g·L$^{-1}$ | 3.22 | Jasova (Slovakia) |

*2.3. Methods Used to Measure the Individual Parameters*

To provide complete results, the basic parameters of the must and wine need to be managed. This mainly concerns the titratable acids, the determination of the sugar content of the must, the assimilable nitrogen, and the alcohol in the resulting wine.

2.3.1. Total Titratable Acids

The titratable acidity of the must was measured using a TITROLINE EASY (SI Analytics GmbH, Mainz, Germany) automatic titrator. A 0.1 mol.L$^{-1}$ solution of sodium hydroxide (NaOH) was used as the titration reagent. For the analyses, 10 mL samples of wine diluted with 10 mL of distilled water were used. Individual samples were then titrated to pH 8.1, with the pH measurement made using a SenTix 21 pH electrode (Fisher Scientific, Loughborough, UK). After titration, the amount of NaOH solution consumed was read off the titrator display. This consumption was multiplied by the NaOH solution factor used in the titration with a coefficient of 0.75. The result gives the titratable acidity of the wine sample (g·L$^{-1}$).

2.3.2. Determination of Sugars (HPLC)

The samples were processed in the laboratories of the Faculty of Horticulture in Lednice (Mendel University in Brno, Czech Republic). The sugars were determined using an HPLC (Shimadzu, Tokyo, Japan) with a refractive index detector. The must and wine samples were centrifuged (3000× *g*; 6 min) and diluted 10 times with demineralized water. Instrumentation: Shimadzu LC-10A binary high-pressure system. System controller: SCL-10Avp, 2 pumps: LC-10Advp, Rheodyne column thermostat with manual injection valve: CTO-10Acvp, DAD detector: SPD-M10Avp.

Column: Watrex Polymer IEX H form 10 μm (Watrex, Praha, Czech Republic); 250 × 8 mm + 10 × 8 mm, separation temperature: 60 °C, sample injection volume: 20 μL.
Mobile phase flow rate: 0.75 mL/min
Isocratic elution
Mobile phase: 1.8 mM H$_2$SO$_4$
Detection: 190 mn

2.3.3. Determination of Alcohol by Distillation

A Gibertini still (Gibertini, Milano, Italy) was used to distil the alcohol. The density of the distillate was obtained through measurement using a Gibertini hydrostatic balance and the alcohol content was obtained by automatic data transfer in DOS.

*2.4. Determination of the Quantity of Volatile Compounds in the Condensate*

Our attention was mainly focused on the most abundant aromatic compounds. This is particularly important from the point of view of the future use of the condensate. The most abundant aromatic compounds were isobutyl alcohol, isoamyl acetate, isoamyl alcohol, 1-propanol, 1-hexanol, ethyl hexanoate, 1-hexyl acetate, ethyl acetate, ethyl decanoate, ethyl acetate, and octanoic acid.

The sample was manually dispensed into the GC system using an injection syringe over six seconds in splitless mode.

Instrumentation: Shimadzu GC-17A, Detector: QP-5050A, Software: GCsolution (https://mygcsolutions.com).

Separation conditions: Column: DB-WAX 30 m × 0.25 mm (Agilent Technologiesm, Santa Clara, CA, USA); 0.25 μm stationary phase (polyethylene glycol); gaseous sample injection volume: 2.5 mL; sampling period 0.2 min (min); carrier gas flow (Helia): 1 mL/min (linear gas velocity 36 cm/s); injection chamber temperature: 200 °C; the initial column space temperature of 35 °C was maintained for four min, followed by a temperature increase gradient of 15 °C/min up to a temperature of 200 °C; total analysis time was 15 min; the detector was operated in SCAN mode, with an interval of 0.25 s in the range of 14–200, to determine the majority of analytes; selective monitoring of ions with fragments 31, 34, 41, 43, 55, 56, 70, 88, 101; detector voltage 1.5 kV; the individual substances were identified on the basis of the MS spectra and retention time.

### 2.5. Statistical Evaluation

The results of the measurement process were processed using Microsoft Excel (Microsoft 365®) and were subject to statistical evaluation using the Statistica 12 (StatSoft CR s.r.o., Prague, Czech Republic) software. Analysis of variance (ANOVA) and factorial ANOVA were used to compare the different categories of the samples. The ANOVA method, using post hoc tests calculated after the primary analysis, allows pairwise comparisons of the mean values of different groups of the samples.

## 3. Results and Discussion

The results from each experiment were divided into two parts. In the first part, attention is paid to the evolution of the condensate composition during the fermentation of the Riesling variety. In the second part, a comparison of the different varieties (Riesling, Merlot, and Sauvignon blanc) and the overall composition of these condensates was made. Our attention was focused on an analysis of the esters and higher alcohols. That is to say, the substances that carry the volatile profile.

### 3.1. Evolution of the Condensate Composition during Fermentation

The Riesling variety was used in the analysis of the evolution of the individual volatile compounds during fermentation (Figure 2). 48 h after the commencement of fermentation with yeast, a collection tank was connected to the condensate collection system and this tank was changed every 24 h. A total of 6 collections was made. Our focus in this case was mainly on the positive aromatics such as isoamyl alcohol, isobutyl alcohol, 1-propanol, ethyl acetate, ethyl hexanoate, ethyl octanoate, and ethyl decanoate. Substances which are not positive are also produced by fermentation, such as acetaldehyde, acetic acid, or acetoin. Last but not least, we also focused on changes in the alcohol content of the condensate with respect to the fermentation phase.

The evolution of the volatiles during fermentation suggests that the most dominant volatile compound in the condensate is isoamyl alcohol. The value on the fifth day reached 1267 mg·L$^{-1}$. This is confirmed by research by Muller [13] where the evolution of aromatics in fermentation gas during fermentation was followed. They also reported that isoamyl alcohol was the most abundant volatile measured. In our research, the second highest levels were found for isobutyl alcohol and ethyl acetate. The value on the fifth day reached 710 mg·L$^{-1}$ and 519 mg·L$^{-1}$. The lowest levels of volatiles were measured for ethyl hexanoate and isoamyl acetate. Isoamyl acetate was not included on the graph due to its low concentration (below the detection limit).

In regard to the development of aromatics with respect to the fermentation phase, their concentrations were highest during the fourth and fifth days of fermentation. Similar results were also obtained in previous research when only the fermentation gases were collected. In that case, the gases were only captured during the main fermentation phase but the highest concentration was measured 144 h after inoculation [14].

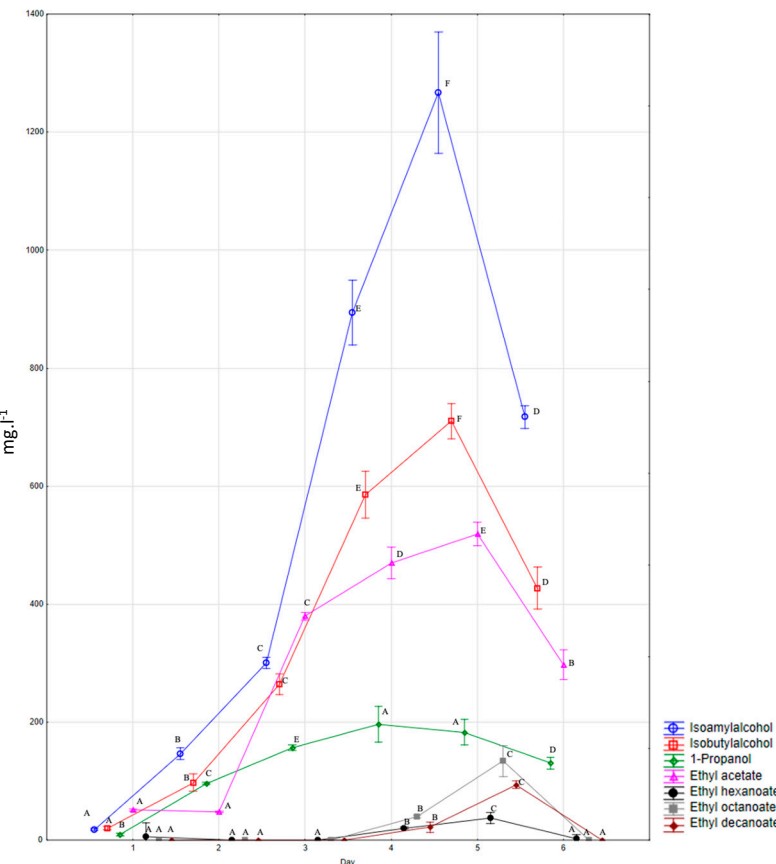

**Figure 2.** The evolution of selected volatile substances in the condensate during fermentation of the Riesling variety. Based on the least significant difference (LSD) test, the results were divided into homogeneous groups. The groups were (A, B, C, D, E, F) and the significance level $\alpha$ = 0.005.

Morakul et al. [15] investigated the quantity of volatiles lost during fermentation. They used a fermentation temperature of 20 °C. Their results found that significant amounts of volatiles are lost during fermentation; up to 56% of the ethyl hexanoate and 34% of the isoamyl acetate can be found in the fermentation gases.

3.1.1. Generation of Undesirable Volatile Components during Fermentation

In our measurement of the development of volatile substances, we also focused on undesirable substances, both in the wine and the fermentation gases, or rather the condensate, such as acetic acid, acetaldehyde, and acetoin.

Figure 3 shows the levels of undesirable volatile substances measured in the condensate; these are considered to have a negative effect on wine. The highest concentration of acetic acid was found in the condensate at the beginning of fermentation 1140 mg·L$^{-1}$, and as fermentation continued the level dropped. These results are also supported by the work of Alexandre [16]. Acetic acid is produced as a by-product of alcoholic fermentation and is mainly formed at the beginning of fermentation [16,17]. At the beginning of fermentation, acetaldehyde was at its minimum level which then increased slightly during fermentation. It reached a peak on the third 333 mg·L$^{-1}$ and fourth day 289 mg·L$^{-1}$ of fermentation. On the following days, its concentration in the condensate decreased. Acetoin was also found to have its minimum concentration at the beginning of fermentation: it peaked on day 2.136 mg·L$^{-1}$ and then decreased as fermentation continued.

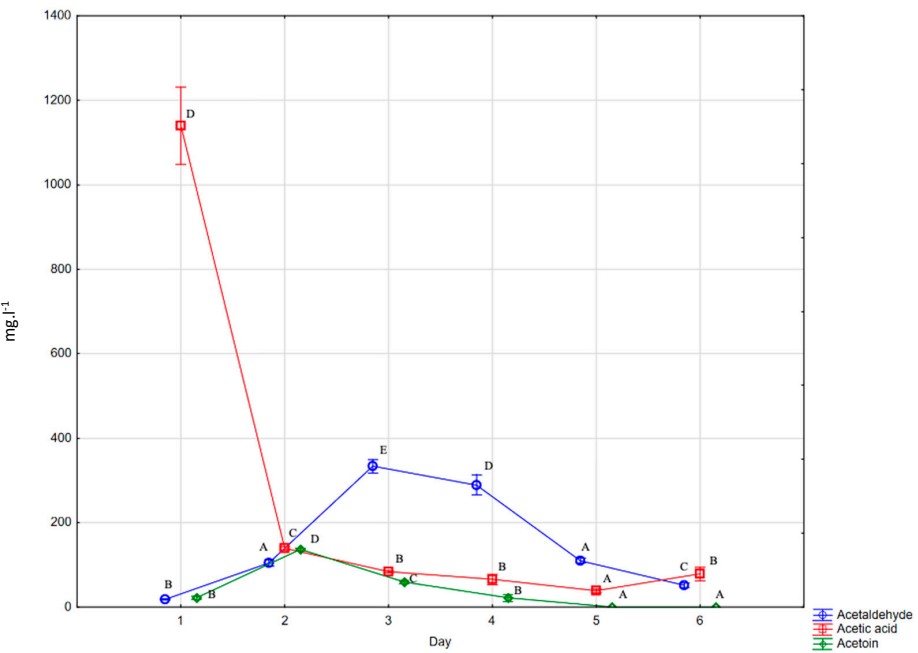

**Figure 3.** The evolution of the undesirable volatiles in the condensate during fermentation. Based on the least significant difference (LSD) test, the results were divided into homogeneous groups. The groups were (A, B, C, D, E) and the significance level $\alpha = 0.005$.

### 3.1.2. Evolution of Ethanol Content during Fermentation

In addition to the development of volatiles in the condensate, the alcohol development was also monitored. For the first 5 days, the ethanol level increased, maximum at 52.47% (Figure 4). This is due to the increasing level of ethanol in the fermentation medium and thus, more ethanol was entrained during fermentation. On the last day of fermentation, the ethanol concentration dropped; this was due to a decrease in the fermentation kinetics and hence less ethanol entered the fermentation gases. According to Luong et al. [18] the fermentation kinetics decrease towards the end of the fermentation due to the high ethanol content of the medium and a reduction in the level of carbohydrates available to sustain the yeast.

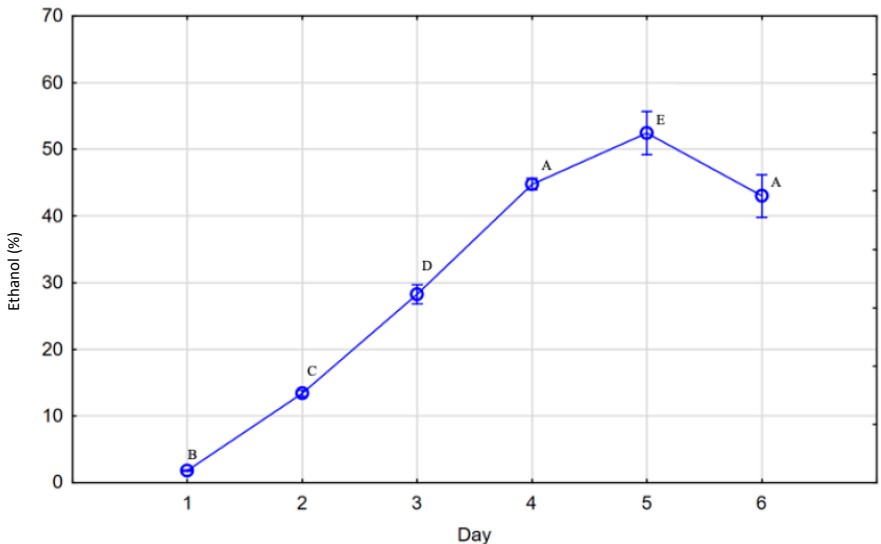

**Figure 4.** The ethanol evolution in the condensate during fermentation. Based on the least significant difference (LSD) test, the results were divided into homogeneous groups. The groups were (A, B, C, D, E) and the significance level $\alpha = 0.005$.

*3.2. Analysis of the Volatiles in the Condensates of the Different Varieties*

In this experiment, the level of the volatiles was only measured at the end of the experiment. All of the condensate, for each individual variety, was collected in a single vessel for the whole period of fermentation and, subsequently, the collected condensate was analysed.

Sauvignon blanc is a variety which typically expresses an aroma of green and tropical notes. This is mainly due to aromatic substances from the methoxypyrazine group and volatile thiols. However, substances from the ester or higher alcohol groups, on which attention has been focused in this study, also play a very important complementary role [19,20].

The most dominant substance that creates the flavour found in the wine was isoamyl alcohol. It is responsible for the banana aroma and belongs to the group of higher alcohols. According to Figure 5, we can say that there is a higher concentration of higher alcohols than esters in the condensate. As far as the esters are concerned, the most strongly represented were ethyl acetate 429 mg·L$^{-1}$ and isoamyl acetate 167 mg·L$^{-1}$. According to a study by Swiegers et al. [21], isoamyl acetate is one of the most important esters in wine. It contributes to the positive aromatic expression of a wine. In contrast, the substance with the lowest concentration, measured in the condensate, was octanoic acid.

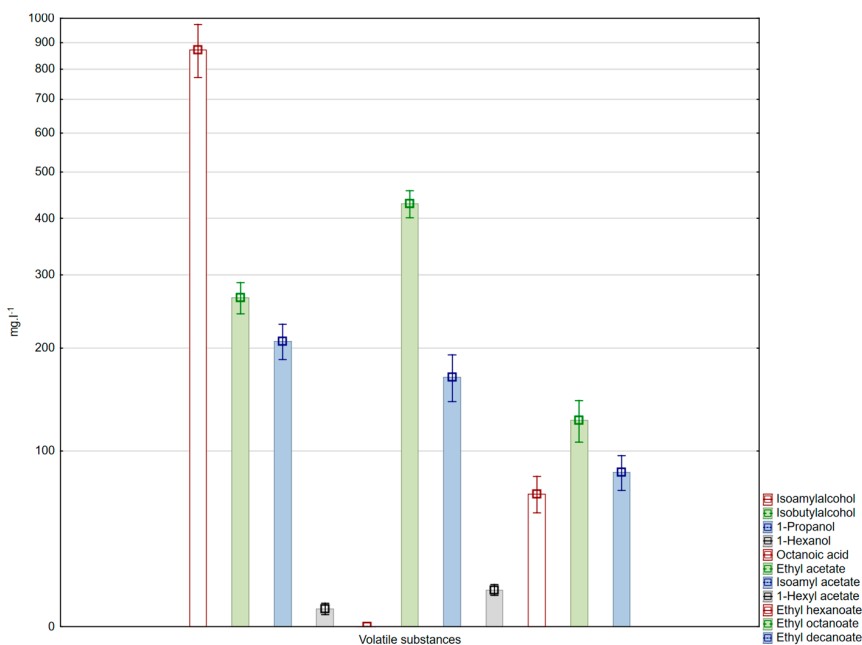

**Figure 5.** The aromatic profile of the condensate from the Sauvignon blanc variety.

According to a study by Suklje et al. [22], the typical aromatic substances found in Riesling are thiols, higher alcohols such as 3-methyl butanol, and esters such as isoamyl acetate or ethyl octanoate (Figure 6) [23]. Within the condensate collected during the fermentation of Riesling, the higher alcohols were the most dominant. The highest levels were found for isoamyl alcohol 463 mg·L$^{-1}$ and isobutyl alcohol 176 mg·L$^{-1}$. It is also interesting that in the condensate ethyl acetate 67 mg·L$^{-1}$ had the highest concentration from the ester group. The other esters were below the limit of detection. Also, in one of the experiments on the Krstač variety, volatile substances from the group of higher alcohols, such as isoamyl alcohol, were predominantly represented [24].

This can be compared to research carried out on fermentation gases from the fermentation of Sauvignon blanc, where isoamyl acetate was found to be the most highly represented. The second most highly represented alcohol was isoamyl alcohol. However, in that experiment it was the gas that escaped during fermentation that was measured and the fermentation temperature was 16 °C [14].

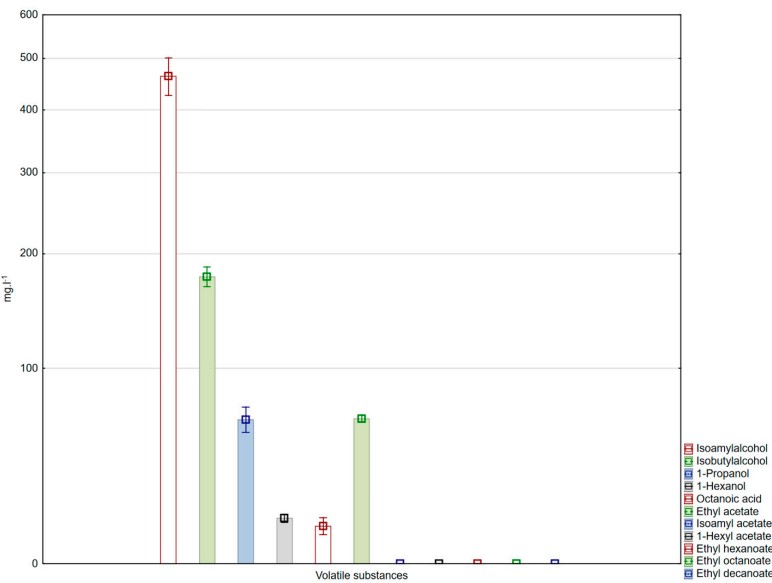

**Figure 6.** The aromatic profile of the condensate from the Riesling variety.

For comparison with the white varieties, the experiment was also carried out using the Merlot variety (Figure 7). The results were similar to those of the Riesling variety. The most highly represented higher alcohol was isoamyl alcohol 803 mg·L$^{-1}$. As far as the esters are concerned, it was ethyl acetate 51 mg·L$^{-1}$. The other esters were found to be below the detection limit. In 2016, an experiment was carried out to capture the condensates from fermentation gases. It was aimed at a total analysis of the condensates but used different grape varieties. According to their results, the total concentration of aromatics in the condensate could reach 1200 mg·L$^{-1}$. The volatile substances with the highest concentrations were propanol, ethyl octanoate, ethyl acetate, and 2,3-butanediol. Isoamyl alcohol was not measured in their study [25].

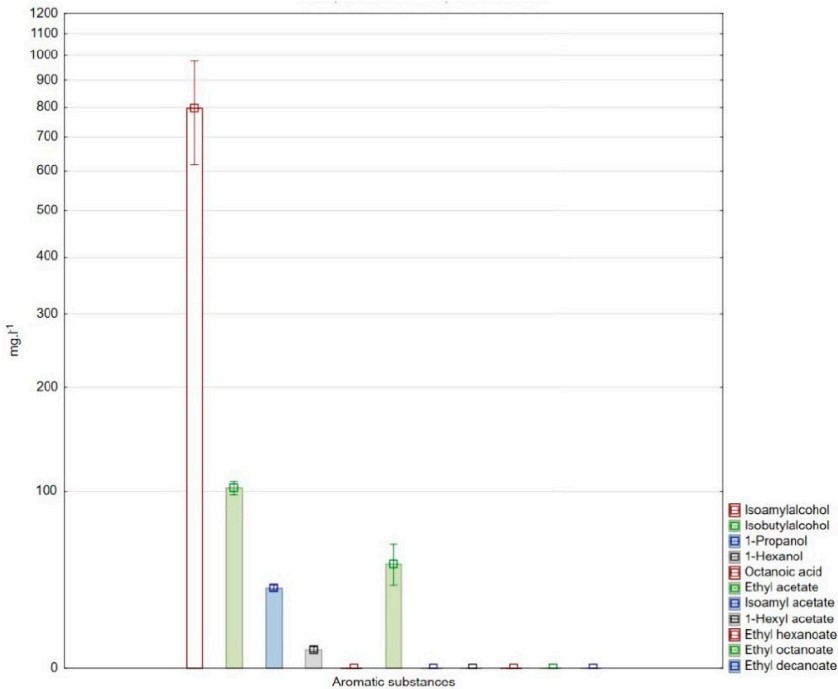

**Figure 7.** The aromatic profile of condensate from the Merlot variant.

In Figure 8, it is possible to see the individual alcohol concentrations found in the condensates from the Riesling, Merlot, and Sauvignon blanc varieties. The Sauvignon blanc variety had the highest alcohol content at 44.87%. The higher alcohol content is due to the higher initial sugar content of the must, which was 25.5 °Bx. Towards the end of fermentation in particular, the fermentation medium had a higher alcohol level and therefore more alcohol was released and subsequently condensed. The Merlot 26% and Riesling 21% varieties had a lower alcohol content and very similar levels of sugar in their respective musts. Gueririni et al. [25] reported an alcohol content in their experiment of 24% vol.; this corresponds well to the results of this study.

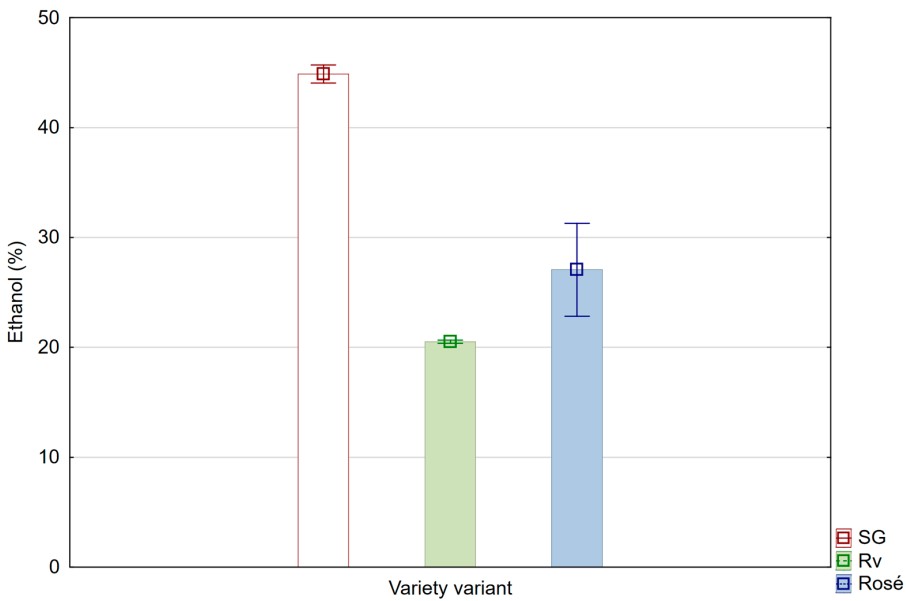

**Figure 8.** Comparison of the ethanol concentration of the different varieties.

## 4. Conclusions

In this experiment, the condensate from the fermentation gases was captured. The results of the study proved that the condensate captured contained a high concentration of volatiles. These are mainly substances from the group of higher alcohols and esters. The results of the evolution of aromatics in the condensate during fermentation showed that the highest concentration of aromatics was found during the fourth and fifth days of fermentation. In the case of the most represented isoamyl alcohol, it was up to 1267 mg·L$^{-1}$ Acetaldehyde was at its lowest at the start of fermentation and at its highest during the second and third days. The alcohol concentration of the condensates varied with respect to the initial sugar content of the must. During fermentation, the highest alcohol content was found at around the fifth day of fermentation. Through a comparison of the composition of the aromatic profiles of the condensate for the different varieties, we can say that Sauvignon blanc has the most complex composition of volatiles. Rosé and Riesling produce similar expressions. A difference is that with the Riesling variety, octanoic acid was also measured. The individual results show that the content of beneficial substances in the condensate is high, and it could be of practical use in winemaking. It might also be used in another food sector as a rich source of aromatic substances or to improve the organoleptic properties of other products within the wine industry.

**Author Contributions:** Conceptualization—J.H., J.S. and M.B.; writing—original draft preparation, J.H., J.S. and P.P.; writing—review and editing, P.P. and M.B.; visualization, J.H.; methodology—M.B. and J.H.; formal analysis—M.K. All authors have read and agreed to the published version of the manuscript.

**Funding:** This paper was supported by the following projects: IGA-ZF/2024-SI1-007, "Study of the condensates of fermentation gases formed during the fermentation of grape must" and by the project CZ.02.1.01/0.0/0.0/16_017/0002334 Research Infrastructure for Young Scientists; this is co-financed by Operational Program Research, Development and Education.

**Institutional Review Board Statement:** Not applicable.

**Informed Consent Statement:** Not applicable.

**Data Availability Statement:** All related data and methods are presented in this paper. Additional inquiries should be addressed to the corresponding author.

**Conflicts of Interest:** The authors declare no conflict of interest.

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
