# Peer review of "A Study of Condensates Collected during the Fermentation of Grape Must"

_fermentation, doi:10.3390/fermentation10040206_

Round 1
Reviewer 1 Report
Comments and Suggestions for Authors
Abstract and the whole manuscript: Please revise the writing style of Merlot rosé. Authors have different writing styles for Merlot rosé, authors should decide if it’s Merlot, Merlot Rose or Merlot rosé.
Line 5: Please delete a space between „acetoin.” and „In addition”.
Table 1: Please revise table, change the header of the table (example: First column: Physicochemical parameters – Sugar, Titrable acids, pH; second column: Riesling – 22 °BX ….). Recommendation: village name can be moved to the title of table: Riesling variety (Dubník, Slovakia) parameters.
Table 2: Please revise table, change the header of the table (example: First column: Physicochemical parameters – Sugar, Titrable acids, pH, village).
2.3.2. Determination of Sugars (HPLC): Please indicate column, injection volume.
Line 153: minutes or min.? Please uniformize the style.
2.5. Statistical evaluation: Please write the manufacturer of the software/city/country.
Line 179: Please write names of volatile substances with lowercase.
Line 269: Please replace 16 C ° with 16 °C.
Please assure that all figures are cited in the text of manuscript; 2.1. Condensation – please cite Figure 1 in the text; same for other figures too. Moreover, assure that order of the figures is correct because authors have 2 figures of figure 1 in the manuscript (page 2 and page 5 have figure 1).
References: Please revise the reference list, most of references are no longer current.
Comments on the Quality of English LanguageModerate editing of English language required
Author Response
Dear reviewwer,
Thank you for your revision. We are sending to you a response on your questions.
Please see the attachment

Reviewer 2 Report
Comments and Suggestions for Authors
Dear Authors,
This is very promising study which is dealing with investigation of condensates profile collected during the fermentation of grape must.
The manuscript is general and in manuscript missing the most important results. Insert values of results in manuscript.
In the section 2 are missing important information regarding grapes which was used in this study.
What was geographical origin of grape? Highlight coordinates and insert it in manuscript.
Which techniques were applied during the growing of grape? Insert it in manuscript.
What kind of soil was in vineyards from which were obtained grapes? Insert it in manuscript.
Did you use any compounds for protection against pests? Insert it in manuscript.
What were climate conditions during vegetative period of grapes in these two regions? Insert it in manuscript.
What was health condition of grapes? Were grapes affected by Botrytis cinerea? Highlight it in manuscript.
Did you add potassium metha bisulphite in grape must during the fermentation?
In the subsection 2.3.2 did you use any external standards for determination of sugars. In the subsection 2.3.2 are missing conditions for HPLC analysis. Insert it.
Did you conduct determination of volatile acids?
In the subsection 3.2. highlight that in analysis of volatile compounds profile of white wine made of Krstač varitey isoamyl alcohol was predominant. Kindly consider to cite J. Serb. Chem. Soc. Vol. 88 No. 1 (2022) 11-23.
Make conslusion more direct. Do not repeat whic you before mantioned in the discussion. Highgiht the most imortant findings.
Wish you all the best in the futre work,
Author Response
Dear reviewer,
Thank you for your revision. We are sending to you a response on your questions.
Please see the attachment

Round 2
Reviewer 1 Report
Comments and Suggestions for Authors
Abstract: Line 11: Please remove space between Merlot and ,; Line 14: Please replace 1-Propanol with 1-propanol;.
Lines 170, 172: Please uniformize the style for minutes, replace with min.
Same suggestion as previous report, please revise list of references, most of references are no longer current.
Author Response
Dear Reviewer,
Please see the attachment. Respond is in section Round 2

Reviewer 2 Report
Comments and Suggestions for Authors
Dear Authors,
Thank you very meuch for revised version of manuscript and answers on my questions and suggestions. It is fine for me.
I made typing mistake in my review in the first suggestion. I meant that abstract is general and that in abstract you have to insert the most important results.
In the manuscript you have interesting results and insert the most important findings in the abstract which will make this paper more recognizable.
Wish you all the best in the future work,
Author Response
Hello,
Thank you for your comments. I have added some of the data from the results to the abstract.
Best regards